# Photoelectric Performance of Two-Dimensional n-MoS_2_ Nanosheets/p-Heavily Boron-Doped Diamond Heterojunction at High Temperature

**DOI:** 10.3390/ijms26104551

**Published:** 2025-05-09

**Authors:** Deyu Shen, Changxing Li, Dandan Sang, Shunhao Ge, Qinglin Wang, Dao Xiao

**Affiliations:** School of Physics Science and Information Technology, Liaocheng University, Liaocheng 252000, China; 2022406623@stu.lcu.edu.cn (D.S.); 17853058528@163.com (C.L.); gsh113026@163.com (S.G.); wangqinglin@lcu.edu.cn (Q.W.)

**Keywords:** n-MoS_2_ NSs/p-DBDD heterojunction, photoluminescence, high temperature, electrical transport behavior

## Abstract

Two-dimensional (2D) n-MoS_2_ nanosheets (NSs) synthesized via the sol–gel method were deposited onto p-type heavily boron-doped diamond (BDD) film to form a n-MoS_2_/p-degenerated BDD (DBDD) heterojunction device. The PL emission results for the heterojunction suggest strong potential for applications using yellow-light-emitting optoelectronic devices. From room temperature (RT) to 180 °C, the heterojunction exhibits typical rectification characteristics with good results for thermal stability, rectification ratio, forward current decrease, and reverse current increase. Compared with the n-MoS_2_/p-lightly B-doped (non-degenerate) diamond heterojunction, the heterojunction demonstrates a significant improvement in both its rectification ratio and ideal factor. At 100 °C, the rectification ratio reaches the maximum value and is considered an ideal high temperature for achieving optimal heterojunction performance. When the temperature exceeds 140 °C, the heterojunction transforms into the Zener diode. The heterojunction’s electrical temperature dependence is due to the Fermi level shifting resulting in the weakening of the carrier interband tunneling injection. The n-MoS_2_ NSs/p-DBDD heterojunction will broaden future research application prospects in the field of high-temperature consumption in future optoelectronic devices.

## 1. Introduction

The transition metal disulfide structure MoS_2_ is one of the typical channel materials used for the semiconductor p–n junction due to its atomically thin n-type properties [1]. It can be referred to as single-layer or multi-layer, corresponding to its number of layers, and the external electric field can be adjusted from the direct band gap semiconductor (1.9 eV) to the natural indirect band gap (1.2 eV), allowing the tunable and large-scale applications of its electronic and optoelectronic properties [2]. With the increasing demand for nano-materials in optoelectronic devices, nanosized MoS_2_ appears to solve various problems encountered in developing excellent optoelectronic devices [3,4], such as transistors [5], sensors [6], LEDs [7], and photodetectors [8,9], because of its good mechanical properties, high carrier mobility, high absorption coefficient and strong electron hole constraints. MoS_2_ has become a top choice to replace previous materials as we enter the new development stage of the nano-era [10]. However, heterostructures formed through different morphologies of MoS_2_ nanostructures (such as 2D thin films and nanosheets), which have been combined with various substrates including MoSe_2_ [11], WSe_2_ [12,13], GaN [14], GaAsSb [15], BA_2_PbBr_4_ [16] and MoGe_2_N_4_ [17] in recent decades, are mostly confined to room-temperature practical applications. Working in the harsh environments of the heterojunction (high temperature, strong radiation, and high pressure), they still face challenges such as device failure and unsatisfactory performance [18]. Boron-doped diamond (BDD) is usually designed and applied in new high-quality high-temperature resistant optoelectronic devices due to its wide band gap, strong chemical stability, and high thermal conductivity in a p-type semiconductor [19]. At present, typical n-type nanostructured semiconductors (such as ZnO [20], Ga_2_O_3_ [21], and WO_3_ [22]) have been used to form p–n heterojunction devices with p-type diamond to construct high-quality new diode terminal extension devices. Recently, we investigated the optoelectrical carrier transport behavior of an n-MoS_2_/p-light B-doped (non-degenerate) diamond heterojunction, which has typical rectification characteristics when working in high-temperature environments [23]. Nevertheless, p-diamond has degenerative characteristics and likely generates band-to-band tunneling through heavily doping boron elements, causing its valence band to lie below the Fermi level [24]. This significantly affects the electrical properties of heterojunctions when combined with other n-type semiconductors [25,26]. Considering the possibility of carrier transport behavior change caused by a heterojunction composed of n-MoS_2_ and p-DBDD, in this innovative research study, a 2D n-MoS_2_ NSs/p-DBDD film heterojunction was designed, and the optoelectrical carrier transport performance was analyzed in depth. Compared with MoS_2_/Si devices [26] and type-II p-MoS_2_/n-InSe vdWs heterojunctions [27,28], the n-MoS_2_ NSs/p-DBDD heterojunction demonstrates superior performance in both its maximum rectification ratio and turn-on voltage, while also exhibiting enhanced high-temperature stability [29]. From room temperature (25 °C, RT) to 180 °C, the heterojunction demonstrates representative rectification properties with excellent thermal stability. The rectification ratio and forward current decrease, and the reverse current increases. The rectification ratio reaches its maximum at 100 °C. Compared with the n-MoS_2_/p-lightly B-doped (non-degenerate) diamond heterojunction in our previous work, this heterojunction demonstrates a significant improvement in both its rectification ratio and ideality factor. The fabricated n-MoS_2_ NSs/p-DBDD heterojunction demonstrates superior current density and evolves into a Zener diode with increasing temperature, while the n-MoS_2_ NSs/p-lightly BDD heterojunction transitions into the reverse diode with increasing temperature. The optoelectrical transport behavior depending on temperature was analyzed through a combined energy band diagram and semiconductor theoretical model, which provides a new direction for high-quality research and development in n-MoS_2_ NSs-related heterojunction optoelectronic devices in harsh environments.

## 2. Results and Discussion

Figure 1a–c displays the morphological characteristics of the 2D n-MoS_2_ NSs with a size range of 1 to 3 μm. Compared to the p-light B-doped diamond film without twinning and with smoother crystallographic facets described in our previous work, the p-degenerated B-doped diamond films exhibit obvious twinned crystal characteristics. The twinned crystal and abundant grain boundaries of p-degenerated BDD films are beneficial in the nucleation of MoS_2_ NSs. The MoS_2_ NSs are densely and randomly distributed on p-DBDD film with a lateral size of about 1.1 μm and a thickness of 98.4 nm. The thickness of 96 nm obtained by local scanning with an atomic force microscope (AFM) confirmed the accuracy of the SEM results shown in Figure 1d,e. The 2D MoS_2_ NSs fixed on p-DBDD film show a uniform coverage and a large leakage boundary and specific surface area, indicating outstanding photoelectric detection advantages [30]. The element-mapping measurement of the MoS_2_ NSs was further confirmed according to the EDS mapping of the spectral area, as shown in Figure 1f. Our observations showed that the elements detected at the heterojunction are C, Mo, and S, as shown in Figure 1g–i which proves that the elements C, Mo, and S coexist and have a high purity. The above characterization proved the successful synthesis of the 2D MoS_2_ NSs on p-DBDD substrates, and no other peaks related to impurities were detected [31].

Figure 2a shows the XRD image of the n-MoS_2_ NSs/p-DBDD heterojunction. In addition to the diffraction peaks of (111) and (220) diamond films at 43.82° and 75.24°, three comparatively sharp diffraction peaks were observed at 14.37°, 61.58° and 68.99°, respectively, which can be attributed to (002), (107), and (201) of 2H-MoS_2_ (JCPD 37-1492). The diffraction peaks at 33.51° and 55.97° are attributed to the presence of (101) and (106) crystal faces in the MoS_2_ structure [32]. The (220) peak of the diamond film is slightly larger than the (111) peak, indicating that the growth rate of the (220) crystal plane of diamond is greater than that of the (111) plane. Li et al. [33] mentioned in their study that the concentration of the B element is closely related to the preferred orientation of diamond during the CVD growth stage. The intensity ratio of (111) and (220) first decreases and then increases with the increase in B concentration, which is consistent with our experimental results. Figure 2b displays the typical Raman image of the n-MoS_2_ NSs/p-DBDD heterojunction. The peak in diamond film at 1330 cm^−1^ comes from the central phonon band, and there is a wider Raman peak at 1520 cm^−1^, meaning that there is a graphite phase in addition to the diamond phase [34]. In order to obtain accurate information about the graphite phase, we fitted the data in the illustration with Lorentz. The Raman peak at 1330 cm^−1^ is in the D-band. Generally, the D-band is more obvious in low-doped diamond films [35]. The Raman peak at 1520 cm^−1^ is in the G-band, which is attributed to the bond stretching of the *sp*^2^ hybrid orbital of heavily boron-doped diamond [36]. The two peaks at 289 and 376 cm^−1^ match the E_1g_ and E^1^_2g_ Raman vibration modes of the 2D n-MoS_2_ NSs [37]. These demonstrate the presence of a 2H-MoS_2_ phase with semiconductor characteristics [38]. In addition, the J1 (154 cm^−1^) band specifies the in-plane shear pattern of one side of the alternating chain relative to the other side [39]. The vibration mode of J2 (241 cm^−1^) is connected with the displacement of the S atomic layer and Mo atoms, while the J3 mode obtained at 336 cm^−1^ includes the stretching of one side of the serrated chain with respect to the other side [40]. All vibration J modes are ascribed to the twisted 1T phase superlattice structure. Therefore, the n-MoS_2_ NSs can reveal both metallic and semiconductor features (2H-MoS_2_ phase) simultaneously.

Figure 2c shows the photoluminescence (PL) spectra excited at 532 nm in the n-MoS_2_ NSs/p-DBDD heterojunction. There is an obvious peak at 568 nm (2.18 eV) due to the transition of exciton A [41,42]. The characteristic broad peak at 660 nm (1.87 eV) is caused by the recombination of the A exciton [43]. In order to further verify the emission color quality of the device, the PL emission CIE color coordinates as shown in Figure 2e, are calculated, and the chromaticity coordinates appear around (0.492, 0.505), which indicates that the n-MoS_2_ NSs/p-DBDD heterojunction shows promise for applications in the research and development of yellow-light-emitting optoelectronic devices. In addition, the electroluminescence (EL) of the device, as observed in Figure 2d, has rich spectral characteristics. In 2D MoS_2_, excitons dominate the absorption and emission characteristics and the recombination process directly related to bound excitons is determined from the angle of EL. MoS_2_ has application value as a tunable portable optical transmitter in which EL is usually confined to the area adjacent to the electrical contact and depends on the hot carrier process [44]. The EL spectrum of the device has three obvious luminescence peaks in the red region. The EL emission peaks at 668 nm and 680 nm are allocated to exciton emission in MoS_2_ [45,46], while the emission peaks at 632 nm are allocated to exciton B emission. As shown in Figure 2e, CIE color coordinates of EL emission data appear near the red emission region (0.309, 0.694). Compared with the n-MoS_2_ NSs/p-non-degenerated BDD heterojunction described in our previous work, the intensity of photoluminescence and electroluminescence is doubled, the PL emission changes from light blue to light yellow, and the EL emission changes from green to red [23].

In order to avoid the short-circuiting of the electrodes, the conductive surface of ITO was used as a conductive cathode in contact with the surface of n-MoS_2_ NSs, and the p-DBDD film was used as a conductive anode with the cathode and anode separated by insulating adhesive. Silver paste and wires were pasted to the conductive side of the ITO conductive glass as the negative electrode. We fixed the p-DBDD film as the positive electrode and the conductive side of the ITOs so that they were in contact with the MoS_2_ nanosheets, to form a pathway. The size of the prepared heterojunction device is 0.25 × 0.25 cm, as shown in Figure 3a. The *I-V* characteristics of the Ag contacts to DBDD and the ITO curve show linear properties indicating an ohmic contact (Figure 3b). The Hall test shows that the p-DBDD has a carrier concentration of 5.8 × 10^21^, a resistivity of 1.05 × 10^−3^, and a mobility of 6.8 cm^2^ V^−1^ s^−1^. Figure 3a–f suggest the electrical properties of the n-MoS_2_ NSs/p-DBDD heterojunction gauged from RT to 180 °C. The *I-V* curve shows good rectification performance for all temperatures. The turn-on voltages are about 0.2 V at RT, 0.7 V at 100 °C, 0.3 V at 120 °C, 1 V at 140 °C, 2.1 V at 160 °C, and 0.7 V at 180 °C. With the temperature increasing, the forward current decreases as the bias increases. At RT, the rectification ratio is about 34.46, and the forward current is 0.16 A at 6 V, which is 34 times higher than the n-MoS_2_ NSs/p-non-degenerated BDD heterojunction reported in our previous work [23]. When the temperature rises to 100 °C, the *I-V* characteristic shows fantastic rectification performance, achieving a rectification ratio of 8.11 × 10^6^ and a decrease in the reverse saturation current of 5.18 × 10^−9^ A. Thus, 100 °C is believed to be the optimal temperature, with excellent performance, for forward rectifier diode n-MoS_2_ NSs/p-DBDD heterojunction. A Zener diode was observed when the temperature reached 140–180 °C. When the breakdown voltage is lower than 4*E_g_*/*q*, the reverse breakdown can be ascribed to the Zener tunneling effect, where *E_g_* and *q* are the semiconductor bandgap and electronic charge, respectively. This device has good application potential for analog circuits and small signal detectors [47]. A noteworthy detail is that the reverse current of the n-MoS_2_ NSs/p-DBDD heterojunction begins to be larger than that at 160 °C and 180 °C, where the rectification ratio is 0.459 and 0.671, and the reverse conduction voltage is 2.1 and 0.7 V, respectively, as shown in Table 1. This is attributed to the increase in reverse carrier tunneling at heterojunctions at high temperatures, while the n-MoS_2_ NSs/p-non-degenerated BDD heterojunction reported in this paper transforms into a reverse diode with increasing temperatures. In addition, the rectification ratio shows a trend of first increasing, then decreasing, and finally increasing again. Due to the thermal excitation effect, the rectification ratio reaches its maximum value of 8.11 × 10^6^ at 100 °C. At 160 °C, due to the enhancement of reverse tunneling, the rectification ratio drops to the minimum value of 0.459. Compared with the n-MoS_2_ NSs/p-non-degraded BDD heterojunction, the n-MoS_2_ NSs/p-DBDD heterojunction has a better rectification ratio at room temperature and high temperatures. At 100 °C, the rectification ratio of n-MoS_2_ NSs/p-DBDD increased by 7.64 × 10^6^ times, which may be due to the addition of B atoms and the high concentration of p-degenerate diamond carriers in heavily doped diamond, resulting in increased carrier injection and tunneling [23].

In order to further explore the electrical transmission behavior at the n-MoS_2_ NSs/p-DBDD heterojunction, Figure 3h,i is combined with the ideal diode in Equations (1) and (2) [48,49]:(1)$$I=I_{s}[\exp(\frac{qV}{nkT})-1],$$
with(2)$$I_{s}=AA^{*}\cdot \exp\left(\frac{-q\Phi_{B}}{kT}\right),$$
where *I_s_* is the reverse saturation current, *q* is the electronic charge, *V* is the applied voltage, *n* is the ideal coefficient, *k* is the Boltzmann constant, *T* is the absolute temperature, *A* is the contact area, and *A** is the effective Richardson constant. On the basis of the 0.1–0.5 V region, the ideality factor (*n* values) of the n-MoS_2_/p-DBDD is 8.98–9.33 from RT to 180 °C, which indicates that the positive values of n-MoS_2_ NSs/p-DBDD are stable. The *n* value is greater than 2, demonstrating that the n-MoS_2_/p-DBDD heterojunction displays nonideal thermal emission properties, which may be due to the inhomogeneity in barrier height, non-uniform interface thickness, and non-uniform atom distribution at the n-MoS_2_ NSs/p-DBDD interface [50,51]. It can be noted that the result for the *n* value is significantly lower than that of the reported n-MoS_2_/p-lightly boron doped BDD heterojunction (8.6–10.2) [52].

Based on the phenomenon of temperature-induced changes in the current and rectification ratio, the equilibrium energy band diagram for n-MoS_2_ NSs/p-DBDD was established, using the Anderson model to explain the carrier migration and barrier variation behavior. Reasonable calculation shows that the conduction band (CB) offset Δ*E*_C_ (3.8 eV) is 10 times higher than the valence band (VB) offset Δ*E*_V_ (0.37 eV), meaning that the injection current is primarily involved in the injected holes in VB. Due to the small VB barrier height at RT, the carrier tunneling from the VB to the defect level band stemming from the MoS_2_ NSs surface state is occupied and exhibits a relatively higher current. Compared with p-non-degenerated BDD, the *E_F_* may approach the band edge and enter into the VB of diamond. The carriers can tunnel from the VB of diamond to the CB of MoS_2_ when forward bias is exerted and show a reduced valence band barrier, higher forward current, and lower diode conduction voltage (Figure 4a,b). In contrast with p-non-degenerated BDD, the Fermi level of which resides within the forbidden band (exhibiting a diffusion-dominated current), the Fermi level of the p-DBDD resides within the valence band (exhibiting tunneling-dominated current) at room temperature. Thus, this shows a relatively high current performance compared to the p-non-degenerated BDD in our previous work. At the high temperature of 120 °C, the barrier height of the VB offset increases, which makes it difficult for carriers to transition from VB to the defect energy band, and the tunneling current gradually disappears. At this time, it is transformed into an ordinary p–n junction diode and shows a low forward current and turn-on voltage (Figure 4c). At 180 °C (Figure 4d), the p-DBDD and n-MoS_2_ thermally activate more charge carriers. In addition, due to the presence of a large number of free surface states and a larger surface-area-to-volume ratio in MoS_2_ NSs, n-MoS_2_ NSs form near-degenerate semiconductors that primarily determine electrical properties [53,54]. The E_F_ may enter the valence band of n-MoS_2_ and the conduction band of p-DBDD near the edge of the band. This induces an increased valence band barrier height. When a bias is applied, as the tunneling current increases, carriers possibly tunnel from the valence band of n-MoS_2_ to the conduction band of p-DBDD with the type of Zener diode (Figure 3f).

The tunneling probability can be expressed using the following simplified relationship:(3)$$T_{B}=\exp(-2\times \frac{\sqrt{2m\Delta V}}{\hbar }\times w_{B})$$
where T_B_ represents the tunneling probability, ΔV denotes the barrier height, and $w_{B}$ corresponds to the barrier width. This relationship indicates an inverse proportionality between the tunneling probability and both barrier dimensions: higher barriers (increased ΔV) and wider barriers (larger W_B_) result in a reduced tunneling probability. When the temperature rises, the n-MoS_2_/p-DBDD heterojunction exhibits a decreased barrier height and narrowed barrier width, leading to enhanced tunneling probability and consequently an increased tunneling current. The experimental current-voltage characteristics provide direct evidence supporting this theoretical framework [55].

The carrier migration behavior can also be explained via the variation of work function, as calibrated in Figure 4. The work function was computed using the following equation:(4)$$\phi =E_{\mathrm{vac}}-E_{F}$$
where *E_vac_* is the energy of the vacuum level [56]. Work function values are indicative of the thermodynamic stability of electrons in the respective components of heterostructures, due to which, upon making contact in the heterostructures, the migration of electrons occurs, due to the electron chemical potential difference at the interface, until reaching fermi-level equilibration. The difference in the work function of the heterojunction with the change in temperature leads to the rearrangement of the electron density at the heterojunction cross-section and generates an induced electric field, which promotes the separation and transfer of electrons and holes. With the increase in temperature, the work function of heterojunction p-diamond decreases, and the work function of n-MoS_2_ increases, meaning that the interface barrier of the heterojunction decreases and more carrier injection is generated at high temperatures, leading to an increase in the tunneling current [57].

The curves at all temperatures in Figure 5a,b are divided into three regions to explore the current transfer mechanism. For low forward voltages (region I), the electrical properties at 25 °C, 100 °C, 120 °C, 140 °C, 160 °C, and 180 °C are directly correlated with the temperature and comply with the power laws of *I*-*V*^1.18^, *I*-*V*^1.22^, *I*-*V*^1.06^, *I*-*V*^1.45^, *I*-*V*^1.14^, and *I*-*V*^1.09^, respectively. The *I*-*V* characteristics of the n-MoS_2_ NSs/p-DBDD heterojunction follow the ohmic laws with a linear relationship at a low voltage at all temperatures [58]. In region II, the heterojunction generally shows the *I*-exp(*V*) relationship attributed to the existence of a composite tunneling performance due to the wide bandgap semiconductors of MoS_2_ and diamond material. By fitting the curve of region II, the injection efficiency constant values were calculated to be 0.63, 0.86, 0.56, 0.67, 1.32, and 0.74 at 25 °C, 100 °C, 120 °C, 140 °C, 160 °C, and 180 °C, respectively [59]. At 160 °C, the exponent value was closest to the standard vacuum diode value (1.5). At the higher voltage region III, the current transport properties comply with the laws of *I*-*V*^1.81^, *I*-*V*^1.72^, *I*-*V*^1.14^, *I*-*V*^3.61^, *I*-*V*^2.21^, and *I*-*V*^1.75^ of 25 °C, 100 °C, 120 °C, 140 °C, 160 °C, and 180 °C, respectively. Current has an exponential relationship with voltage, and the exponents are greater than 2, which is normally attributed to the conduction model of space charge limiting current (SCLC) transmission model, which indicates that the SCLC mechanism is limited by a single dominant trap level [60]. The SCLC model with restricted traps can account for exponential values close to 2. At high temperatures, the number of excited holes increases, accumulating and expanding the space charge region within the junction, and, according to the SCLC model, the current mechanism changes from J∝V^2^ to J∝V^3/2^, and thus the exponent is raised to a value greater than 2. At the higher temperature of 180 °C, the exponent falls down again, exhibiting an almost-linear relationship, close to the ohmic–voltage–current relationship, because fewer holes can be excited for injection from diamond VB to MoS_2_ VB due to the higher energy levels, and this mainly occurs through the tunneling effect [61].

The interface carrier transmission performance at the n-MoS_2_ NSs/p-DBDD heterojunction can be investigated by fitting the plots of ln(*I/V^2^*) versus 1/*V* into three situations (Figure 5c,d). (I) For low bias voltage at high temperatures, multiple carriers pass through the interface with the energy produced by thermal excitation to overcome the interface barrier. The carriers transport mechanism can be illustrated using Equation (5) to interpret thermionic emission:(5)$$I=AA^{*}\exp[\frac{-(\emptyset_{b}-\sqrt{\frac{q^{3}V}{4\pi \varepsilon_{0}\varepsilon_{r}d}})}{KT}],$$
where *A* is the area of the heterojunction, *A** is the Richardson constant, *d* is the height of the potential barrier at the interface, $\emptyset_{b}$ is the height of the potential barrier at *T* = 0 K, *ε*_0_ is the dielectric constant of the vacuum, and *ε_r_* is the dielectric constant of the semiconductor. Thus, the capability of thermionic emission to conquer the interface barrier height is dependent on the demand excitation temperature. As the temperature is insufficient to allow carriers to pass through the interface barrier, the carrier transfer is primarily concerned with the interface barrier tunneling mechanism, which is identical to the energy band diagram illustration. (II) The interfacial carrier transfer mechanism for low bias voltage can be expounded by coupling the direct tunneling and energy band diagram (Equation (6)):(6)$$\ln\left(\frac{I}{V^{2}}\right)\propto \ln\left(\frac{1}{V}\right)-\frac{4\pi d\sqrt{2m\emptyset_{b}}}{h},$$
where *h* is Planck’s constant, and *m* is the charge carrier effective mass. (III) For the higher voltage, the carrier transport mechanism primarily involves Fowler–Nordheim (F–N) tunneling and is shown in Equation (7):(7)$$\ln\left(\frac{1}{V^{2}}\right)\propto -\frac{1}{V}(\frac{8\pi d\sqrt{2m\emptyset_{b}^{3}}}{h}),$$

Therefore, multiple carriers can acquire enough energy to pass through the thermionic emission barrier at high temperatures. Based on bias voltage, the carrier injection could take place in direct tunneling to F–N tunneling at lower temperatures. In addition, the emergence of the inflection point (*V_t_*) at RT–140 °C indirectly illustrates the presence of F–N and direct tunneling. As *V_tRT_* > 6.25 V, *V_t_*_100_ > 2.3 V, *V_t_*_100_
*>* 2.3 V, *V_t_*_140_ > 4.6 V, *V_t_*_160_ > 2 V and *V_t_*_180_ > 1.40 V, 1/V exhibits a negative slope trend, implying that the emergence of the F–N tunneling effect on the carrier electrical transport behavior varied, and 1/V took a logarithmic form, signifying the appearance of a direct tunneling effect. With the F–N tunneling mechanism governing carrier transport, the n-MoS_2_ NSs/p-non-degenerated BDD heterojunction exhibits a higher turn-on voltage (*V_tRT_* = 7.69 V, *V_t_*_100_ = 6.67 V, *V_t_*_120_ = 3.84 V, and *V_t_*_140_ = 5.56 V), whereas the n-MoS_2_ NSs/p-DBDD heterojunction demonstrates significantly reduced thresholds (*V_tRT_* = 6.25 V, *V_t_*_100_ = 2.3 V, *V_t_*_140_ = 4.6 V, *V_t_*_160_ = 2 V and *V_t_*_180_ = 1.40 V). Furthermore, compared with the MoS_2_ NSs/p-non-degenerated BDD heterojunction, the n-MoS_2_ NSs/p-DBDD heterojunction requires a smaller inflection point voltage and depends more on the F–N tunneling conduction mechanism. The lower inflection point voltage at 160 °C and 180 °C means that the heterojunction is more dependent on the F–N tunneling mechanism, which offers a new perspective on erasing electrons in the field of memory devices. In addition, the phenomenon of a large reverse current occurring when a large voltage is applied at high temperatures can be attributed to thermionic activated carriers and an amplified tunneling current [62,63].

## 3. Materials and Methods

In an instrument filled with flowing H_2_ and CH_4_ gases, heavily boron-doped p-type diamond films were prepared on a silicon wafer substrate by the hot-filament chemical vapor deposition (HFCVD) method. The boron source was provided by liquid trimethyl borate ((CH_3_O)_3_B), which was introduced into the chamber through H_2,_ and the DBDD membrane was obtained by controlling the flow rate of H_2_. Subsequently, the DBDD membrane was washed with ethanol and then deionized water to remove residual impurities from the surface. The DBDD was not treated with irradiation, acid-boiled, or treated with a high temperature, so it had a hydrogen surface terminal.

MoS_2_ nanosheets were prepared on DBDD film via the sol–gel method. As shown in Figure 1, [(NH_4_)_6_Mo_7_O_24_ · 4H_2_O] was used as the Mo source, CH_3_CSNH_2_ as the S source, and C_14_H_23_N_3_O_10_ as the chelating agent. The mixture was stirred continuously for one hour using a magnetic stirrer and dissolved in 8 mL of deionized water to obtain a sol. We placed the DBDD film based on silicon at the center of the spin-coating machine, used a pipette to drop the sol onto the center of the DBDD film, and accelerated from 0 to 3000 rpm within 56 s. We removed the film and set it on a heating table at 60 °C to cure the coating on the DBDD film for 5 min, and then we performed the second deposition through an identical procedure to thicken the nanosheet with the annealing process at 400 °C over 4 h to improve the quality of the bonded device. Figure 2a illustrates the schematic structure of the n-MoS_2_ NSs/p-DBDD heterojunction. The MoS_2_ NSs are brought into contact with the conductive side of transparent indium tin oxide (ITO) glass and secured using cyanoacrylate adhesive to prevent direct contact between the conductive copper wires and TiO_2_. The BDD at the bottom of the contact can easily cause a short circuit, which can be effectively prevented by employing ITO as a dielectric. The ITO and conductive copper wires are connected with silver paste, respectively, to fabricate the heterojunction anode and cathode. The linear *I*-*V* characteristics between the ITO/Ag and BDD/Ag contacts demonstrate the linear relationship of an ohmic contact (Figure 2b).

A scanning electron microscope with energy dispersive X-ray spectroscopy (SEM, Thermo Fisher Scientific FIB-SEM GX4, produced from Bothell, WA, USA) was used to examine the element distribution and structural morphology of the samples. We set the scanning speed of the X-ray diffractometer (XRD, D8 ADVANCE, produced from Karlsruhe, Germany) at 5°/min to check the phase structure and purity of the sample. The molecular structures of MoS_2_ NSs and p-degenerated BDD were detected with an excitation wavelength of 532 nm using a Raman spectrometer (Raman, inVia, produced from London, UK), and the heterojunction electrical properties were tested with the Keithley 2400 source (Keithley Instrument, Cleveland, OH, USA). The carrier concentration of the DBDD film was assessed via the Hall effect (ET9110-HS, produced from Beijing, China).

## 4. Conclusions

In conclusion, an n-MoS_2_ NS/p-DBDD heterojunction was successfully fabricated with obvious twin characteristics and abundant grain boundaries on p-DBDD films, which made the MoS_2_ NSs easier to nucleate. The *I-V* characteristics of the devices all show rectification characteristics at RT-180 °C, and the rectification ratio reaches the maximum at 100 °C, which is considered the optimal rectification temperature for the heterojunction. In contrast to the n-MoS_2_ NS/p-lightly BDD heterojunction, which transitions into a reverse diode at elevated temperatures, the fabricated n-MoS_2_ NS/p-DBDD heterojunction exhibits Zener diode characteristics when operating above 140 °C. A Zener diode was observed when the temperature was above 140 °C. The variation in the electrical transport behavior at different temperatures can mainly be attributed to the shift in the Fermi level and the change in the tunneling current. The rectification ratio and ideal factor of n-MoS_2_ NS/p-DBDD are significantly superior to those of the prepared n-MoS_2_ NS/p-non-degenerated BDD heterojunction. A noteworthy detail is that the n-MoS_2_ NS/p-DBDD heterojunction is more dependent on F–N tunneling and is suitable for operation under high-temperature conditions. The n-MoS_2_/p-DBDD heterojunction provides an effective reference for applications of memory erasure technology and the development of high-quality devices for use in extreme working environments.

## Data Availability

Data is contained within the article.

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
