# Peer review of "Photoelectric Performance of Two-Dimensional n-MoS2 Nanosheets/p-Heavily Boron-Doped Diamond Heterojunction at High Temperature"

_ijms, 2025, doi:10.3390/ijms26104551_

Round 1
Reviewer 1 Report (Previous Reviewer 1)
Comments and Suggestions for Authors
Review of the manuscript ijms-3591876 - Revised Manuscript for the Authors: In the second revised version, the authors have clearly distinguished their current research from their previous work. The manuscript is now ready for publication.
Author Response
We deeply appreciate the reviewers' acknowledgment of the scientific value and methodological rigor demonstrated in this study.
Reviewer 2 Report (New Reviewer)
Comments and Suggestions for Authors
This study investigates the high-temperature optoelectronic behavior of n-MoS2 nanosheets/p-DBDD heterojunctions, demonstrating promising rectification ratios and Zener diode characteristics. While the experimental design and thermal stability results are noteworthy, the manuscript requires enhanced mechanistic analysis and methodological clarity.
1. Lack of Contextual Novelty
- Issue: Insufficient comparison with recent MoS2/diamond studies (e.g., 2023 reports on interfacial doping effects).
- Suggestion: Benchmark against similar heterojunctions (e.g., MoS2/SiC) to emphasize the role of heavy boron doping.
2. Incomplete Data Interpretation
- Issue: High ideality factors (n > 8) attributed to “non-uniform barriers” without experimental proof (e.g., C-V profiling).
- Suggestion: Perform C-V measurements to quantify interface states or cite analogous works (e.g., Xie et al., 1996).
3. Mechanism Clarification
- Issue: Qualitative band diagrams lack quantitative support for Zener tunneling (e.g., tunneling probability calculations).
- Suggestion: Use Anderson model to simulate temperature-dependent barrier modulation and carrier injection.
Comments on the Quality of English Language
- Grammar: Correct typos (e.g., “posemetakaolin” → “metakaolin”) and ambiguous phrases (e.g., “graphite particle size below 1500” → specify units).
- Figures: Improve AFM resolution (Fig. 1e) and label axes clearly in logI-logV plots (Fig. 5a-b).
Round 2
Reviewer 2 Report (New Reviewer)
Comments and Suggestions for Authors
the manuscript is now in satisfactory condition.
This manuscript is a resubmission of an earlier submission. The following is a list of the peer review reports and author responses from that submission.
Round 1
Reviewer 1 Report
Comments and Suggestions for Authors
Review of the manuscript ijms-3524450-peer-review-v1 for the Authors: This article presents research on the Photoelectric performance of two-dimensional n-MoS2 2 nanosheets/p-heavily boron-doped diamond heterojunction. The first major objection I have is the similarity of the this work with this: https://doi.org/10.1016/j.jallcom.2023.172819
In principle you prepared exactly the same samples, did almost the same characterizations, so I am struggling to find the novelty in your research. Other comments are listed below.
1) Introduction needs to be broaden a bit, maybe add some references for the heterojunction behavior.
2) Scheme 1 needs to be upgraded.
3) Figure 1 also needs to be upgraded, SEM is too small, and why did you choose these magnifications.
4) AFM measurements, why did you choose only this part of the AFM scan for the cross section?
5) “three 125 comparatively sharp diffraction peaks were observed at 14.37°, 61.58° and 68.99° respec-126 tively, which can be attributed to (100) of 2H-MoS2 (JCPD 37-1492). The diffraction peaks 127 at 33.51° and 55.97° are attributed to the presence of (101) and (106) crystal faces of the 128 MoS2 structure [22].” Can you explain this sentence, are all of these peaks from MoS2 or not?
6) “All vibration J modes are ascribed to the twisted 1T phase 148 superlattice structure.” of MoS2?
7) PL spectra look somehow weird, was the data smoothed or?
8) The IV characterization, and the extensive discussion about the mechanism is the best part of the manuscript.
9) Conclusions and references are ok.
In conclusion, since the paper lacks novelty, I cannot recommend it until the authors address this issue. Therefore, my current recommendation is to reject it.
Reviewer 2 Report
Comments and Suggestions for Authors
The manuscript ijms-3524450 reports the fabrication of Two-dimensional (2D) n-MoS2 nanosheet (NSs) was deposited on p-type heavily boron-doped diamond (BDD) film to form n-MoS2/p-degenerated BDD heterojunction device. The heterojunction electrical temperature dependence is due to the Fermi level shifting resulting in the weakening of the carrier interband tunneling injection. The n-MoS2 NSs/p-DBDD heterojunction will broaden the subsequent research application prospects in the field of high-temperature consumption of future optoelectronic devices. The proposed work highlights the importance of p-n junctioned heterointerfaces and their photoelectric properties at different temperatures. Overall, the present manuscript is interesting in the field of optoelectronic devices and suitable for publication after the following revisions.
- How the authors confirmed the nature of MoS2 as N-type and BDD as p-type.
- I appreciate the authors for providing the detailed characterization of n-MoS2/p-degenerated BDD heterojunction devices.
- I advise the authors provide a detailed charge transfer mechanism based on the workfunctions of both MoS2 and BDD.
Round 2
Reviewer 1 Report
Comments and Suggestions for Authors
Review of the manuscript ijms-3524450-peer-review-v2 for the Authors: In the revised version, the authors have addressed almost all of my comments. However, my main concern remains unresolved. The authors have not sufficiently explained the differences between their paper and the following study: https://doi.org/10.1016/j.jallcom.2023.172819. The similarities between the two papers are too significant, and the submitted work does not demonstrate enough novelty. If the authors can clearly differentiate their study and highlight its unique contributions, I will reconsider my decision. However, for now, my recommendation remains to reject the paper.
